# Mechanical Properties, Melting and Crystallization Behaviors, and Morphology of Carbon Nanotubes/Continuous Carbon Fiber Reinforced Polyethylene Terephthalate Composites

**DOI:** 10.3390/polym14142892

**Published:** 2022-07-16

**Authors:** Liang Qiao, Xu Yan, Hongsheng Tan, Shuhua Dong, Guannan Ju, Hongwang Shen, Zhaoying Ren

**Affiliations:** School of Materials Science and Engineering, Shandong University of Technology, Zibo 255000, China; qiaol66@163.com (L.Q.); polaris_xu@foxmail.com (X.Y.); dongshuhua@sdut.edu.cn (S.D.); ju@sdut.edu.cn (G.J.); shwhappylife@163.com (H.S.); rzy0709@163.com (Z.R.)

**Keywords:** poly(ethylene terephthalate), carbon nanotube, continuous carbon fiber, dynamic mechanical analysis, differential scanning calorimetry, crystallization kinetics

## Abstract

Carbon nanotube/continuous carbon fiber reinforced poly(ethylene terephthalate) (CNT/CCF/PET) composites are prepared by melt impregnating. The effects of CF and CNT content on the mechanical properties, melt and crystallization behaviors, and submicroscopic morphology of CNT/CCF/PET composites are studied. The tensile test results show that the increase of CF and the addition of appropriate amount of CNT improved the tensile strength and tensile modulus of the composites. When the content of CNT is 1.0 wt% and the content of CF is 56 wt%, the properties of the composites are the best, with tensile strength of 1728.7 MPa and tensile modulus of 25.1 GPa, which is much higher than that of traditional resin matrix composites. The results of dynamic mechanical analysis (DMA) show that the storage modulus of the composites increased with the increase of CF and CNT content. In particular, the addition of CNT greatly reduced the loss modulus of the composites. Morphological analysis show that the addition of CNT improved the fiber–matrix interface of the composite, which changes from fiber pull-out and fracture failure to fiber matrix fracture failure, and the fiber matrix interface is firmly bonded. In addition, there are polymer coated CNT protrusions on the surface of the fiber was observed. The results of differential scanning calorimetry (DSC) show that the melting temperature and crystallization temperature of the composites increased with the increase of CF content. The addition of CNT had little effect on the melting temperature of the composites, but it further improved the crystallization temperature of the composites. The effect of CNT content on the crystallization kinetics of the composites is studied. The non-isothermal crystallization kinetics of the composites is described by Jeziorny’s improved Avrami equation. The results show that CNT has a great influence on the crystallization type of the composites. As a nucleating agent, CNT has obvious heterogeneous nucleation effect in the composites, which improves the crystallization rate of PET.

## 1. Introduction

Continuous carbon fiber reinforced thermoplastic (cCFRTP) composites are widely used in automotive industry, aerospace, military industry, and other fields because of their light weight, high strength, and high toughness [1,2]. It is of great significance to the lightweight process in various fields and has great market application space. The main difference between cCFRTP composites and traditional thermoplastic composites is that the fibers of the former exist continuously in the whole material system, while the fibers of the latter are discontinuous. Therefore, the mechanical properties of cCFRTP composites are better than those of traditional discontinuous fiber reinforced composites [3], and the application prospect is wider. The continuous fiber characteristics of cCFRTP composites determine that they are highly anisotropic, so it allows various directional performance reinforcement designs. At present, melt impregnation is the most mature and widely used preparation process for preparing cCFRTP composites, with high preparation efficiency and low cost. However, this process has short impregnation time and high viscosity of polymer melt, resulting in poor infiltration of fiber matrix interface. The most common way to improve the interfacial properties of cCFRTP is to add toughening filler to the resin. Wang et al. [4] grafted MWCNT and GNP on the fiber surface to prepare carbon fiber reinforced resin composites. It was found that the interface between fiber and matrix was improved and the mechanical properties of the composites were improved. Zhang et al. [5] prepared carbon nanotube/carbon fiber reinforced polyimide (CNT/CF/PI) composites by melt impregnation method. The effect of CNT on the fiber–matrix interface was studied. It was found that the mechanical properties of CNT/CF/PI composites were better than those of CF/PI composites. Despite research progress, the incorporation of carbon nanomaterials into composites are highly required, yet rather challenging. Meanwhile, there are still few studies on the properties of CNT and fiber–reinforced resin matrix composites.

Poly(ethylene terephthalate) (PET) is selected as the resin matrix. It is the most common and widely used thermoplastic resin in life. It is an attractive high-performance polymer with sufficient mechanical and thermal properties [6,7] and a wide range of applications. However, as a typical semi crystalline resin, it has slow crystallization speed and low melt strength [8,9]. Therefore, they are generally not suitable for manufacturing extruded profiles, pipes, or structural parts. The physical and mechanical properties of semi crystalline polymers are strongly affected by their microstructures, and the crystallization behavior will affect the formation of their microstructures. So far, many studies have introduced nano fillers [10,11,12,13,14,15] (for example, carbon nanotubes, carbon black, graphene, clay) to prepare nanocomposites to improve some properties of PET, such as improving crystallinity, conductivity, and interface properties. Among these nano fillers, carbon nanotubes are the most promising field in the research of nanocomposites, which have been widely used and added to polymer matrix [16]. Carbon nanotubes have significant mechanical and electrical capabilities. Their unique structure and high aspect ratio make it possible to significantly improve the crystallization ability, mechanical properties, and electrical conductivity of polymers with only a small amount of addition. At the same time, they have high cost-effectiveness. Khalid et al. [17] prepared CNT-embedded high flexibility sensor by three roll milling method. The structure shows that the sensor has good electrical stability and high sensitivity. Jang et al. [18] realized the effective dispersion of CNT and polymer through the proposed new preparation method of polymer sensor. The results show that the sensor has high flexibility (100%), good resistance variation (160%), and repeatability (r^2^ > 0.98). Khalid et al. [19] prepared CNT polymer sandwich composites by spraying. The results show that this method is simple and efficient, CNT can be effectively dispersed, and CNT content can be controlled by controlling the concentration of CNT solution. Gao et al. [20] prepared functional multi wall carbon nanotubes (f-MWCNTs)/PET nanocomposites by melt blending. The results showed that f- MWCNTS had obvious nucleation effect on PET crystallization and had a great influence on PET crystallization behavior. Antoniadis et al. [21] prepared MWCNTs /PET nanocomposites with different contents of MWCNTs by in situ polymerization. The thermal properties and morphology were analyzed. The results show that MWNTs have a positive effect on the crystallization rate, crystallinity, and morphology. Then, the various preparation methods reported in these studies have particular advantages and disadvantages. Among them, the melt mixing method is simple, fast, and can achieve mass preparation, which matches the preparation process of the prepreg in this study. Therefore, the melt mixing method was used to realize the effective dispersion and mixing of CNT and polymer matrix.

cCFRTP composites have attracted extensive attention because of their excellent properties. It is of great significance to prepare higher performance composites and explore the effects of various factors on their properties for expanding their engineering applications. In this study, carbon nanotube/continuous carbon fiber reinforced polyethylene terephthalate (CNT/CCF/PET) composites are prepared by melt impregnation method. A series of technical problems such as low carbon fiber content, uneven distribution, and poor impregnation effect are effectively solved by using a uniquely designed continuous fiber melt impregnation mold. The purpose of this work is to study the effects of different CF and CNT content on the mechanical properties, melting and crystallization behaviors, and submicroscopic morphology of the composites. The effects of CF and CNT content on the strength and modulus of the composites are studied by tensile test and dynamic mechanical analysis (DMA). The tensile fracture surface of the sample is examined by scanning electron microscope (SEM) image to observe the fiber polymer interface and fracture form. The melting and crystallization behaviors of the composites are studied by differential scanning calorimetry (DSC). The non-isothermal crystallization kinetics of the composites is described by Jeziorny’s improved Avrami equation. The effects of CNT content on the crystallization ability and crystallization rate of the composites were studied.

## 2. Experimental

### 2.1. Materials

CF, T700SC–24k, was produced by Toray Co., Ltd., Tokyo, Japan. The type was. PET, BG–80, was supplied by Sinopec Yizheng Chemical Fiber Co., Ltd., Yizheng, Jiangsu, China. CNT, GT–210, was produced by Shandong Exhibition Nanomaterials Co., Ltd., Binzhou, Shandong, China. The antioxidants, IRGANOX-1010, IRGAAFOS-168, were produced by BASF Gao-Qiao Performance Chemicals (Shanghai) Co., Ltd., Shanghai, China.

### 2.2. Preparation of Prepreg Tapes

#### 2.2.1. CNT/PET Composite Pellets

The process diagram for preparing CNT/PET pellets is shown in Figure 1. PET was dried at 160 °C under vacuum for 6 h. PET, CNT, and antioxidant 1010 and 168 with various ratios are mixed first in high speed mixer for 3 min (the mixer was produced by Dongguan Huanxin Machinery Co., Ltd., Guangdong, China) and then extruded into pellets via a single screw extruder produced by Harbin Hapu Electric Technology Co., Ltd.,Harbin, China. The extrusion temperature of the barrel was 270 °C. The ratios of PET and CNT are listed in Table 1.

#### 2.2.2. Preparation of Prepreg Tape

The CNT/CCF/PET prepreg tapes were prepared by a special continuous fiber reinforced thermoplastic composite melting laboratory impregnation device; it was designed and assembled by the authors. The preparation process diagram is shown in Figure 2. Yarn frame (ADC–2) was produced by Guangzhou Aosai Carbon Fiber Technology Co., Ltd., Guangdong, China. The single screw extruder with a screw diameter of 20 mm and three-roll calendar (CL–T110) were produced by Harbin Hapu Electric Technology Co., Ltd., Harbin, China. 

The CCF/PET pellets were dried at 160 °C under vacuum for 6 h, and were added into the single screw extruder for melting and plasticization. They were extruded into the impregnation mold, and the fiber then passes through the mold through the yarn frame and was dispersed and impregnated therein. If pulled out of the mold, during this period, the polymer molecular chains and carbon fibers will have a high orientation change, shaped and calendered through the three roll calender. CNT/CCF/PET prepreg tapes were then obtained and finally curled into a roll. The formula of the prepreg belt is shown in Table 2. The shape of the prepared partial prepreg tape is shown in Figure 3. The width of the prepreg tape was 14 ± 0.2 mm and the thickness was 0.3 ± 0.01 mm, and it exhibited high toughness.

### 2.3. Measurements

#### 2.3.1. Tensile Strength

The tensile properties were measured with a mechanical properties testing machine (INSTRON 5969, Instron Corporation, Norwood, MA, USA) according to ASTM D3039/D3039 M–14, and the tensile speed was 10 mm/min. The tensile strength of every sample was obtained from the average value of five effective data.

#### 2.3.2. Differential Scanning Calorimetry

The melting and crystallization behavior of pure PET and CNT/CCF/PET composites were measured by using a differential scanning calorimeter (DSC Q2000, TA instruments, New Castle, DE, USA). All operations were performed under a nitrogen atmosphere with a sample weight of about 5 mg. All the samples were first heated from room temperature to 270 °C at a rate of 20°C/min and held at this temperature for 3 min to erase the thermal history. Then, the samples were cooled to 50 °C at different constant rates of 5, 10, 20, and 40 °C/min, respectively. Finally, the samples were also reheated to 270 °C at a rate of 10 °C/min after cooling to 50°C to characterize the melting and crystallization behaviors of the composites.

#### 2.3.3. Dynamic Mechanical Analysis

The dynamic mechanical behavior of CNT/CCF/PET composites was tested by a dynamic mechanical analyzer (DMA Q800, TA instruments, New Castle, DE, USA). The sample size was 20 × 5 mm, and was tested in a tensile mode at a frequency of 1.0 Hz. The test was performed with temperature from −20 °C to 180 °C at a heating rate of 5 °C/min.

#### 2.3.4. Fractural Analysis

The tensile fracture surface was observed by scanning electron microscope (Quanta 250, FEI Company, Hillsboro, OR, USA). The fracture surfaces of the specimens were sputter-coated with Pt.

## 3. Results and Discussion

### 3.1. Mechanical Property

#### 3.1.1. Tensile Property

Figure 4 shows the representative stress–strain curves of tensile samples of CNT/CCF/PET composites with different carbon fiber content and CNT content. The average value and corresponding standard deviation of tensile strength and elastic modulus are shown in Table 3. 

For CCFRT, the matrix plays a role in transmitting load when bearing tensile load, and the fiber bears the main load and is the main contributor of tensile strength and tensile modulus. It can be seen from these results that under the reinforcement of CCF (fiber content 36 wt%), the tensile properties of CCF/PET composites are significantly improved, reaching 998.6 MPa, and with the increase of fiber content (CF content 56 wt%), the tensile properties of CCF/PET composites are increased by 56.9 wt%, reaching 1567.1 MPa, which is higher than that of ultra-high strength steel (1130 MPa). The tensile modulus is also significantly increased to 19.8 GPa, which shows that carbon fiber plays a major role in the tensile properties of CCF/PET composites. High fiber content brings higher strength and modulus, but high fiber content brings higher challenges to the preparation of composites and the effective impregnation belt of fibers. In addition, the high strength of CCF/PET composites shows that the special preparation process of CCFRT increases the molecular orientation of polymer chains, increasing the tensile properties [22]. Sood et al. [23] reported similar conclusions.

CNT, as an ideal filler for polymer, will increase the tensile strength and tensile modulus of polymer, and observe the decrease of elongation at break [24]. The results show that the addition of CNT further improves the tensile strength and tensile modulus of the composites. With the addition of only 0.1 wt% CNT, the tensile strength continues to increase by 6.5% and the tensile modulus increases by 11.2%. When the content of CNT was 1.0 wt%, the tensile strength of the composite reached the maximum value of the experiment, 1728.7 MPa, and the tensile modulus increased significantly to 25.1 GPa. It can be seen that the reinforcing effect of CNT is very significant. The effective dispersion of CNT in the matrix can form a reinforcing network and affect the dominant properties of the matrix and the properties of fiber matrix. Some studies have proved that CNT plays an important role in enhancing the crack propagation ability of polymer matrix due to the uniform interaction between reinforcing material and matrix [25,26,27]. However, it is worth mentioning that the addition proportion of CNT in the composite is further increased (2.0 wt%), providing a significant reduction in strength and modulus. These differences are closely related to the agglomeration of CNT in the matrix and the formation of defects in the composites, which leads to the reduction of the properties of the composites. On the other hand, at the highest CNT concentration (i.e., 2.0 wt%) in the composites, the formation of aggregates is higher, which reduces the interfacial filler polymer adhesion. Valentini et al. [28] proved that the number of aggregates increases with the increase of CNT concentration. Aggregation will affect the insertion of polymer molecules between CNTs, that is, CNTs do not adhere to the matrix, thus affecting the properties of composites.

#### 3.1.2. Dynamic Mechanical Analysis

The storage modulus (*E′*), loss modulus (*E″*), and tan delta (*tanδ*) values of composites are shown in Figure 5a–c respectively. It can be seen from Figure 5a that the *E′* increases with the increase of fiber content. The increase of fiber content allows greater stress transfer between fiber and matrix, resulting in the increase of *E′*. When the movement of polymer segments is limited by fiber–fiber and fiber–matrix interactions, the *E′* of composites will increase [29,30]. It can also be seen that *E′* increases with the addition of CNT. The content of CNT is 1.0 wt%, and *E′* of the composite reaches 234 GPa, which is 40.9% higher than that without CNT. The high *E′* and toughening effect of CNT will act as a stress transfer bridge between fiber and matrix, so as to enhance the stress transfer between fiber and matrix. It can be seen that the *E′* of the composite decreases when the CNT content is 2.0 wt%. This is related to the agglomeration of CNT in the matrix and the formation of defects. It can be seen that the *E′* of the composite decreases with the increase of temperature. At low temperature, the *E′* decreases slowly. When the temperature reaches *T_g_*, the *E′* decreases obviously, which is related to the movement of molecular segments [31]. With the increase of temperature, the matrix and fiber lose freezing and become easy to move, and the stress transfer between matrix and fiber decreases, resulting in a decrease of *E′* [32]. It can also be seen that the *E′* of CCF/PET composites without CNT decreases significantly when the temperature reaches near *T_g_*, while the *E′* of CNT/CCF/PET composites with CNT decreases slowly and can still maintain a high E′ when reaching *T_g_* temperature. The addition of CNT will limit the movement of molecular segments, so as to weaken the effect of stress transfer failure.

As can be seen from Figure 5b, the *E″* of the composite increases with the increase of temperature, and the *E″* with high fiber content increases. The greater the fiber content, the stronger the partition effect of the fiber on the matrix and the greater the *E″*. The maximum *E″* appears near the *T_g_* temperature, after which the *E″* of the composite decreases with the increase of temperature. It can also be seen that after the addition of CNT, the increase of *E″* with the increase of temperature is indeed significantly weakened, and the initial *E″* is low, but the maximum value is still near the *T_g_* temperature. The *E″* indicates the viscoelasticity of the polymer. The more the *E″* is, the closer the material is to the ideal viscoelastic material, which shows that CNT has an obvious toughening effect on the matrix and greatly improves the toughness of the matrix. This improvement is still related to the content of CNT. When the content of CNT reaches 1.0 wt%, the effect is the best, while when the content of CNT reaches 2.0 wt%, the effect is reduced.

As can be seen from Figure 5c, *tanδ* increases with the increase of temperature, and the peak value of CCF/PET composite is significantly larger than that of CNT/CCF/PET composite. *Tanδ* reaches the peak at *T_g_* temperature, indicating that the mechanical loss of the composites is the largest in the glass transition zone. After adding CNT, the *tanδ* of the composite changes gently with temperature, and the peak decreases, indicating that the wider relaxation time distribution of molecular segment motion and the lower *tanδ* indicate higher viscoelastic behavior of the material [33].

### 3.2. Morphology Analysis

The tensile fracture surfaces of CNT/CCF/PET composites specimens with different fiber content and CNT content were conducted using SEM observation, as shown in Figure 6. Figure 6a,b show the tensile fracture of CCF/PET composite with 36 wt% fiber content. It can be widely observed that the fibers were debonded and pulled out, and there was no polymer matrix coating on the fiber surface. The main reason for the interface separation between fiber and matrix is the insufficient interface bonding between fiber and matrix [34], and lack of interaction between the carbon fiber and matrix [35]. In the tensile test, the matrix was broken and the interface stress was insufficient, resulting in fiber pull out and fracture failure [36], which corresponds to low tensile strength. Figure 6c,d shows the tensile fracture of CCF/PET composite with 56 wt% fiber content. It can be observed that part of the fiber was debonded and a part of the fiber surface was still coated with polymer matrix. The enhancement of interfacial bonding effect shows that the increase of fiber content improvs the extrusion impregnation effect of fiber and matrix in a limited die volume, which corresponds to higher tensile strength. Figure 6e,f shows the tensile fracture surface of CCF/PET composite with CNT content of 1.0 wt% and fiber content of 56 wt%. It can be observed that the fiber and the matrix was broken and failed, the polymer matrix with obvious adhesion on the fiber surface, and the interface between the fiber and the matrix was closely bonded, which represented the highest tensile strength. It is worth mentioning that the protruding layer of polymer coated CNT can be observed on the surface of the fiber. Potschke et al. [37] observed thick polymer coated nanotubes protruding from the fiber surface. Coleman et al. [38] observed similar behavior and linked it to the formation of high-strength crystal coatings. Assouline et al. [39] observed the crystallization nucleation of polymer in the presence of nanotubes and we found that its structure is similar to that in this paper, which indicates that the formation of this structure is closely related to the crystallization behavior of CNT in polymer, and this structure can enhance the interface bonding between fiber and matrix. It can be shown that the interfacial strength between fiber and matrix is an important factor to determine the tensile strength. This is because due to the insufficient adhesion strength of the fiber matrix, the matrix can no longer hold the fibers together during the tensile load, and the composite fails prematurely due to the cumulative weakening of the structure [40].

### 3.3. Melting and Crystallization Behavior

#### 3.3.1. Melting

Figure 7a,b show the melting behavior curves of pure PET and all the as-prepared samples at a heating rate of 20 °C/min, and the corresponding melting parameters are shown in Table 4. The relative crystallinity (*X_c_*) was calculated using the equation [41]:(1)$$X_{c}=\frac{∆H_{m}}{\left(1-W_{f}\right)∆H_{0}}$$
where ∆*H_m_* is melting enthalpy during the heating process; ∆*H*_0_ is enthalpy for 100% crystallization PET, which is 140 J/g [42] and *W_f_* is the mass fraction of CCF and CNT.

Figure 7a shows the melting behavior curves of Pure PET and CCF/PET composites with fiber content of 36 wt% and 56 wt%. The results show that compared with pure PET, continuous fiber reinforcement can increase the melting point (*T_m_*) of the composite by 6.5 °C, but higher fiber content does not significantly change the *T_m_*. In fact, the following factors affect the movement of *T_m_* of polymer composites toward high temperature [43,44]. First, the *T_m_* of polymer will be increased if the expansion and free movement of polymer molecular segments are hindered. The closely arranged CF in the prepreg zone will undoubtedly greatly hinder the expansion and free movement of polymer molecular segments, leading to the free movement of polymer molecular segments at higher temperatures, resulting in the increase of *T_m_*. Second, the molecular chain orientation and the external force applied during material processing will also improve the *T_m_* of the polymer. This is related to the processing and preparation process. The CCF/PET prepreg belt is extruded in the die, which will improve the orientation of polymer molecular chains. Moreover, the prepreg belt is rapidly cooled and shaped by the pressing roller, and the material is applied with internal stress, resulting in the increase of *T_m_*. Finally, the *T_m_* of polymer composites is closely related to the crystallization behavior, and the increase of *X_c_* or crystal integrity will increase *T_m_*. The addition of CF increases the crystallinity of the material, resulting in the increase of *T_m_*. For crystallinity, we can see from the Table 4 that *X_c_* of the composite decreases at high fiber content (56 wt%). This is because the spacing between fibers is small at too high CF content, which hinders the diffusion and migration of molecular chains and inhibits crystal growth [45,46].

Figure 7b shows the melting behavior curve of CCF/PET composites with CNT contents of 0.1 wt%, 1.0 wt%, and 2.0 wt%. The results show that for *T_m_*, the content of CNT has no apparent influence on the *T_m_* of the composites. Gao et al. [15] reached the same conclusion in the study of f–MWCNTs/PET nanocomposites. Most scholars have reached similar conclusions on the effect of various nano fillers/PET composites on *T_m_* [47,48,49]. The addition of CNT plays the role of heterogeneous nucleation such as CF, the addition of an appropriate amount of CNT will improve the crystallinity of the composite. It can be seen from Table 4 that the *X_c_* of CNT/CCF/PET with 0.1 wt% is 18.8%, which is 2.3% higher than that without CNT. With the addition of 1.0 wt% CNT, the *X_c_* reached the highest 20.1%, but increased little. Moreover, when the CNT content continued to increase and reached 2.0 wt%, the *X_c_* of the composite decreased significantly, has lost the role of CNT as heterogeneous nucleation site to improve the crystallinity. The effect of CNT addition is closely related to the effective dispersion of CNT in the polymer. The addition of a large amount of CNT cannot effectively disperse in the polymer, but will agglomerate and inhibit the crystallization of PET.

#### 3.3.2. Crystallization

Figure 8 shows the crystallization melting curves of pure PET and CNT/CCF/PET composites with different fiber content and CNT content, and the non-crystallization parameters are listed in Table 5. Figure 8a shows the results of crystallization behavior of pure PET and CCF/PET with fiber content of 36 wt% and 56 wt% from the melt at a cooling rate of 20 °C/min. For pure PET, the melt-crystallization peak temperature (*T_c_*) is 156.5 °C and a broad melt-crystallization peak is observed, which means the melt-crystallization process of pure PET is very slow. As a result, both crystallization rate and nucleation rate are very slow, corresponding to the low crystallization temperature [50]. However, for the CCF/PET composites, the addition of carbon fiber has a great influence on the melt-crystallization behavior of PET. When CF content is 36 wt%, the *T_c_* increases up to 191.5 °C, 35.5 °C higher than that of pure PET. Furthermore, a sharp melt-crystallization peak is observed for the composite. It can also be seen from Figure 8a that with the increase of CF content in composites, the *T_c_* continues to increase, when CF content is 56 wt%, the *T_c_* increases up to 201.3 °C, 9.8 °C higher than that of 36 wt% CCF/PET composites. *T**_c_* moves to high temperature, indicating that CF plays a role in heterogeneous nucleation, promotes the crystallization of the system, and enables PET to start crystallization at a higher temperature. In other words, the addition of CF provides a large number of nucleation sites, the molecular chains are attached to it for orderly arrangement, and induce the growth of the crystalline layer around its surface, so as to nucleate and crystallize at a higher temperature, which corresponds to the shift of *T_o_* and *T_c_* to high temperature in the DSC cooling curve.

Figure 8b shows the results of the crystallization behavior of CCF/PET with fiber content of 56 wt% and CNT content of 0, 0.1 wt%, 1.0 wt%, and 2.0 wt% from the melt at a cooling rate of 20 °C/min. As can be seen from Figure 8b that *T_c_* continues to increase with the increase of CNT in the composite, and it can also be seen that the composite crystallizes in the early stage of the cooling process (i.e., higher temperature), which is because the dispersed CNT acts as a nucleating agent in the composite, which is the same as the heterogeneous nucleation of CF and consistent with the previously reported results [51]. Only a small amount of 0.1 wt% CNT has a great impact on the crystallization behavior of composite. Compared with no CNT, *T_c_* increases by 15.3 °C, and *T_o_* moves to a higher temperature. Compared with the composites with CNT content of 0.1 wt%, the *T_c_* of the composites with 1.0 wt% CNT did not change significantly or even decreased slightly, and the *T_c_* of the composites with 2.0% CNT decreased significantly. This means that for the melting crystallization behavior of PET in composite, there is a saturation of the nucleation effect of CNT [52,53].

### 3.4. The Non-Isothermal Melt Crystallization Kinetics

Avrami equation [54,55,56,57] describes the time evolution of nucleation and growth of crystal domain by using relative crystallinity. It is widely used to determine the crystallization kinetics of various nucleation and growth modes of polymers under isothermal conditions. Jeziorny [58] modified the Avrami equation to consider non isothermal crystallization and the effect of heating/cooling rate on parameter *Z_t_*. In Jeziorny’s analysis, the kinetic constant is determined in the same way as using Avrami equation [59].

Avrami equation assumes that the relative crystallinity (*X_t_*) develops with crystallization time (t/s), as follows
(2)$$1-X_{t}=exp\left(-Z_{t}t^{n}\right)$$
where *n* is Avrami exponent, *X_t_* is relative crystallinity and *Z_t_* is the crystallization rate parameter.

This can be written as
(3)$$\log\left[-\ln\left(1-X_{t}\right)\right]=\log Z_{t}+n\log t$$

For non-isothermal crystallization at certain cooling rate Jeziorny’s method is to modify *Z_t_* by the equation
(4)$$\log Z_{c}=\frac{\left(\log Z_{t}\right)}{\Phi }$$
where *Φ* is the cooling rate, *Z_t_* is the kinetics crystallization rate.

The relative crystallinity (*X_t_*), as a function of temperature is defined as [60]
(5)$$X_{t}=\frac{\int_{T_{o}}^{T}\left(dH_{c}/dT\right)}{\int_{T_{o}}^{T_{\infty }}\left(dH_{c}/dT\right)dT}$$
where *T_o_* and *T_∞_* are the onset and end of crystallization temperature, respectively. *H_c_* is the heat flow at temperature *T*.

The DSC curves of CCF/PET composites with different CNT contents when cooled at different cooling rates are shown in Figure 9. These are used to evaluate the non-isothermal crystallization behavior of the system, and finally to determine the correlation between *X_t_* and *t*, non-isothermal crystallization kinetic parameters and crystallization activation energy [61]. It can be seen that as the cooling rate increases from 5 °C/min to 40 °C/min, the start of melt crystallization temperature (*T_o_*) and crystallization peak (*T_c_*) move to a lower temperature and the peak becomes wider. With the increase of cooling rate, the time for the polymer chain to overcome the nuclear barrier becomes shorter, resulting in crystallization at lower temperature, because the nucleation activity at lower temperature becomes stronger [62,63]. However, the faster cooling rate also leads to the shorter time for macromolecules to diffuse into the microcrystalline lattice and adjust the configuration to a more perfect microcrystalline. Therefore, the faster cooling rate leads to a wider melt crystallization peak [64]. It is worth mentioning that from Figure 9c, we can see that the melting crystallization peak of the prepreg belt has a shoulder peak on the high-temperature side. First of all, the test sample has shoulder peaks at all cooling rates, which indicates that this phenomenon is not accidental, but related to the properties of the material. Then, this sample is added with 1.0 wt% CNT content. As discussed earlier, CNT with 1.0 wt% content has the most obvious effect on crystallization, which indicates that the occurrence of this phenomenon is closely related to the crystallization of CNT. Shoulder peaks appear on the high-temperature side in the melting crystallization process, which is usually considered to be caused by the formation of micro crystallites of different sizes in the primary crystallization process [65,66]. CNT, as an excellent and effective nucleating agent, improves a large number of nucleation sites and promotes pet to nucleate and form microcrystals at higher temperatures, but the crystallization rate of microcrystals decreases. With the decrease of temperature, the microcrystals are transformed into more perfect crystals, and the crystallization rate is faster, so the shoulder peak phenomenon appears.

Figure 10 shows the relationship between the relative crystallinity (*X_t_*) and crystallization time (t/min) of non-isothermal melt-crystallization of CCF/PET composites with different CNT content at different cooling rates. We can see that the crystallization time decreases and the crystallization rate increases with the increase of cooling rate.

The Avrami plots for the non-isothermal melt crystallization of CCF/PET composites with different CNT content is shown in Figure 11. The values of the Avrami exponent *n* and the rate constant *Z_t_* can be determined from the slope and the intercept of the plot of *log[−ln(1−**X_t_**)]* versus *log t*. A series of straight lines were obtained from the plots, which demonstrated the good application of the Avrami equation. The curve of the composite shows the initial linear part, which then tends to be stable. The linear part of the curve is considered to be the result of primary crystallization, and the deviation of the curve is considered to be due to secondary crystallization, which is caused by spherulite impact in the later stage. Table 5 lists the values of *n* and *Z_c_* and half-time of complete crystallization (*t_1/2_*) at different cooling rates. The *n* values were about 3.6 for CCF/PET. This indicates that there is a lazy time-dependent initial nucleation in the composites, and homogeneous nucleation and heterogeneous nucleation exist simultaneously in the crystallization process. From the Table 5, we can find that all samples conform to a law, that is, the *n* value and *Z_c_* value increase with the increase of cooling rate, and the *t_1/2_* value decreases. The increase of *n* value with the cooling rate indicates that the crystallization behavior of PET becomes more complex at high cooling rate, which is closely related to the stronger nucleation activity [67] and shorter diffusion time of molecular chain movement at lower temperature. With the increase of cooling rate, *Z_c_* value increases and *t_1/2_* value decreases, indicating that the sample can crystallize faster at higher cooling rate. It can be seen from Table 6 that the *n* values of the CCF/PET added with 0.1 wt% and 1.0 wt% CNT increases significantly. The variation of *n* values indicates that the existence of CNT as nucleating agent affects the non-isothermal crystallization process, including the nucleation type and spherulite growth of composites. For the CCF/PET with 2.0 wt% CNT, the *n* values did not change significantly compared with the CCF/PET without CNT, indicating that 2.0 wt% CNT did not play the due nucleation effect in the composites. Generally, the occurrence and incompleteness of various crystallization mechanisms and the volume change caused by phase transition will affect the accuracy of Jeziorny equation. In fact, this equation is usually mainly applied to the primary crystallization process [68].

### 3.5. The Non-Isothermal Melt Crystallization Activation Energy

For the non-isothermal melt crystallization process, the differential iso-conversional method developed by Friedman [69] and Viazovkin [70] can be used to evaluate the effective activation energy of polymer crystallization. 

The method is based on the following equation:(6)$$\ln\left(\frac{d\alpha }{dt}\right)=\ln\left(\beta \frac{d\alpha }{dT_{\alpha }}\right)=\ln A+\ln f\left(\alpha \right)-\frac{\Delta E}{RT_{\alpha }}$$
where *α* is the crystallization conversion degree, *β* is the cooling rate, and *f(α)* is the function describing the reaction mechanism, *A* the pre-exponential factor, *R* is the universal gas constant, *T* is the temperature at a given *α*, t is the time, and *E* is the effective activation energy. It is obvious from the equation that for a given *α*, the plot of *ln**[**dα⁄dt]* versus *1/T_α_* obtained from curves recorded at several cooling rates should be a straight line, and its slope gives the value of Δ*E*. 

The Friedman plots of the sample are shown in the Figure 12. The value of Δ*E* can be obtained from the slope of the plots and is listed in Table 7. As can be seen from Table 7, as predicted by L–H theory, the Δ*E* value is negative, which indicates that the crystallization rate increases with the decrease of temperature. It can also be seen that the activation energy of the composite increases monotonically and the Δ*E* value depends on *α*. It shows that the crystallization mechanisms with different activation energies participate in the complex reactions, which occur at different degrees of crystallization conversion. At the same conversion *α*, the Δ*E* value of CCF/PET composites with 0.1 and 1.0 wt% CNT is lower, indicating that its crystallization rate is faster, and CNT does play a role in heterogeneous nucleation. The Δ*E* value increased with the addition of 2.0 wt% CNT, indicating that the effect of CNT on hindering the movement of molecular chain is higher than that of heterogeneous nucleation, which is related to the poor dispersion and agglomeration of CNT, which indicates that an appropriate amount of CNT can be used as an effective nucleating agent for PET.

## 4. Conclusions

The results show that the fiber content and CNT content have a great influence on the mechanical properties, crystallization and melting behavior, and micro morphology of CNT/CCF/PET composites. The tensile test shows that the increase of fiber content and the addition of appropriate amount of CNT improved the tensile strength and tensile modulus of the composites. The properties of CNT/CCF/PET composites are better than those of CCF/PET composites. The tensile strength of CNT/CCF/PET composites (CNT content 1.0 wt%, CF content 56 wt%) reached 1728.7 MPa and the tensile modulus reach 25.1 GPa, which is much higher than that of traditional resin matrix composites.

DMA results show that the storage modulus of the composites is improved with the increase of fiber content and CNT content. However, the loss modulus of CCF/PET composites is high, and the addition of CNT significantly reduced the loss modulus of the composites, indicating that the addition of CNT has an obvious toughening effect on the composites.

Morphological analysis show that the interface adhesion between CCF/PET composite fiber and matrix is poor, the fiber is pulled out from the matrix and breaks, and the fiber surface is smooth without coating the matrix. The addition of CNT effectively improved the fiber matrix interface of the composites. CNT/CCF/PET composite fiber is firmly combined with the matrix. The matrix is coated on the fiber surface and failed due to the fracture between the fiber and the matrix. It is also observed that there is a prominent CNT structure coated with matrix on the surface of the fiber. Strong interface bonding and CNT structure reinforcement are the reasons for the high strength of CNT/CCF/PET composites.

DSC results show that the increase of CF content increased the melting temperature and crystallization temperature of the composites, and the addition of CNT further increased the crystallization temperature of the composites. The study of crystallization kinetics show that the content of CNT has an important influence on the crystallization rate and crystallization type of the composites. The promotion effect of CNT on PET crystallization come from the increase of crystal growth point, which could be used as an effective heterogeneous nucleation agent, but the promotion effect of CNT is closely related to its effective dispersion.

Finally, the study confirm that CCF/PET composites have very excellent properties. Through the synergistic reinforcement and toughening of carbon nanotubes, CNT/CCF/PET composites can be used as high-performance multifunctional materials and play a greater role in the preparation and application of composites.

## Figures and Tables

**Figure 1 polymers-14-02892-f001:**
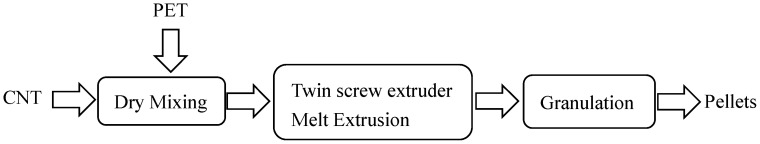
Schematic diagram for the process of CNT/PET pellets.

**Figure 2 polymers-14-02892-f002:**
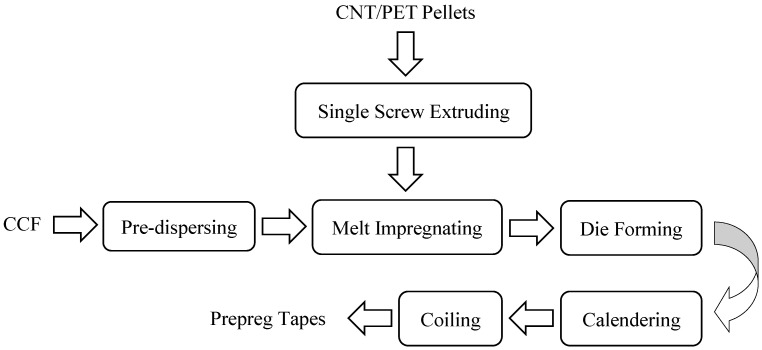
Schematic diagram for process of CNT/CCF/PET prepreg tapes.

**Figure 3 polymers-14-02892-f003:**
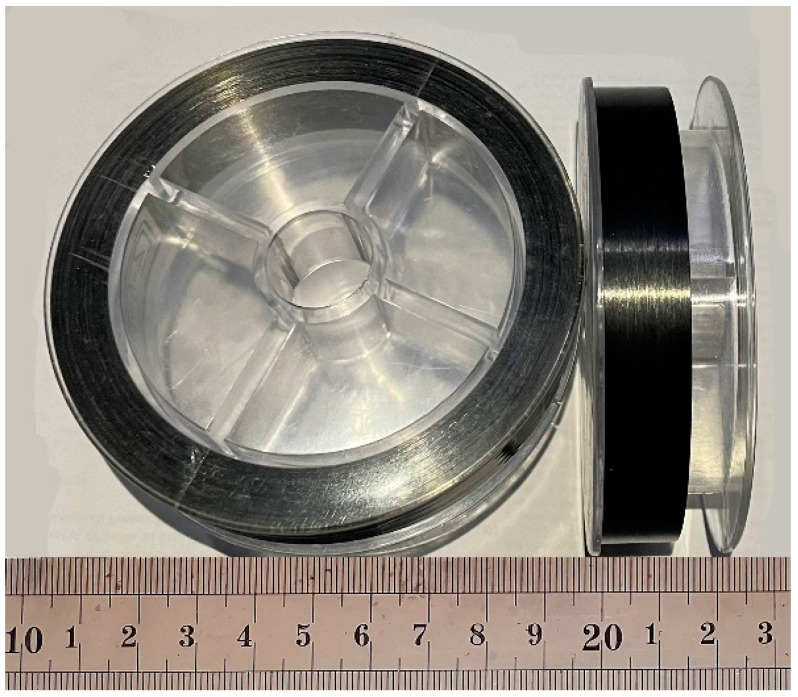
Prepreg tapes of CNT/CCF/PET.

**Figure 4 polymers-14-02892-f004:**
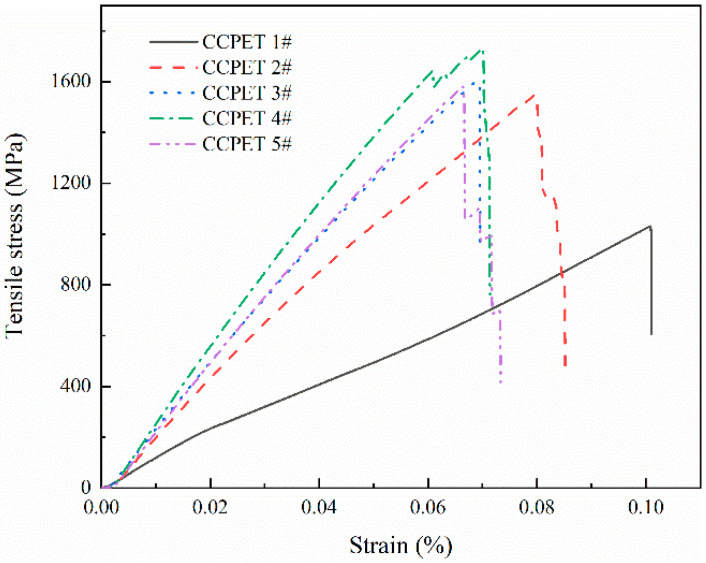
Representative tensile stress–strain curves of tensile specimens of CNT/CCF/PET composites with different fiber content and CNT content.

**Figure 5 polymers-14-02892-f005:**
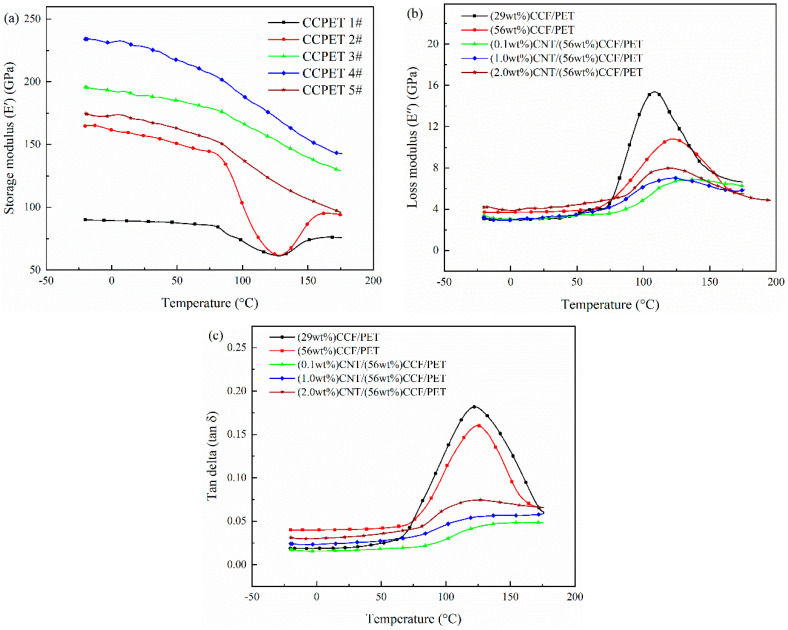
DMA test results of CNT/CCF/PET composites: (**a**) storage modulus, (**b**) loss modulus, (**c**) *tanδ*.

**Figure 6 polymers-14-02892-f006:**
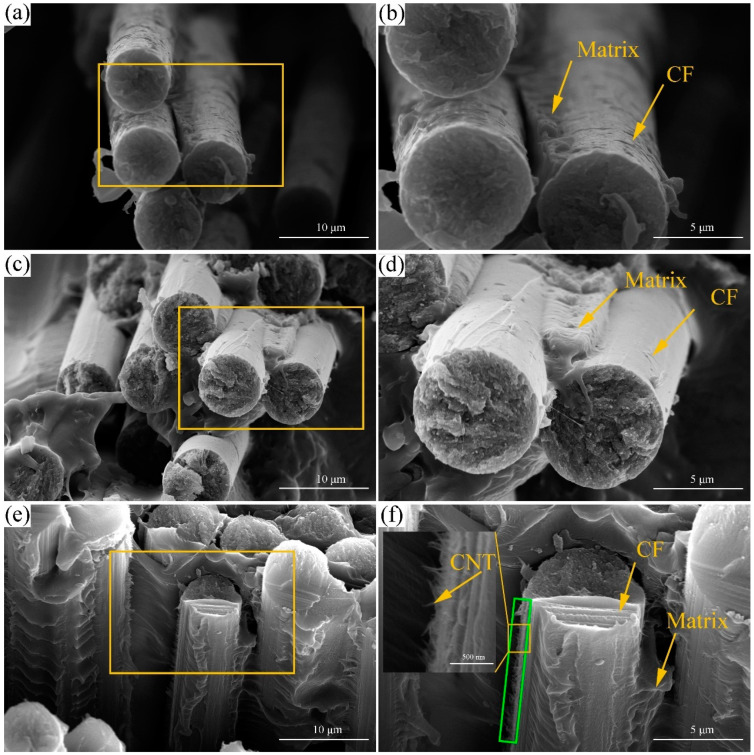
Microstructure of the fracture surface of the tensile specimen of the prepreg tape: (**a**,**b**) (36 wt%) CCF/PET, (**c**,**d**) (56 wt%) CCF/PET, (**e**,**f**) (1.0 wt%) CNT/(56 wt%) CCF/PET.

**Figure 7 polymers-14-02892-f007:**
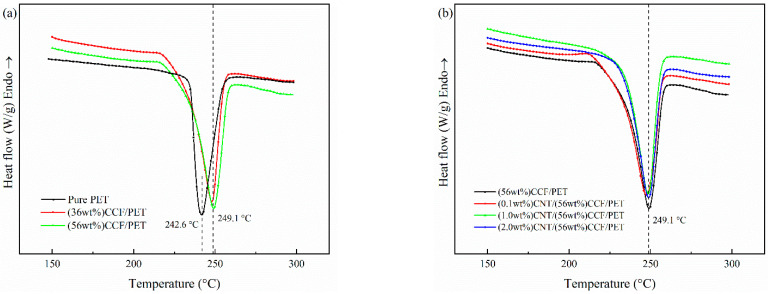
Melting behavior curves of pure PET, CCF/PET, and CNT/CCF/PET composites: (**a**) pure PET, (36 wt%) CCF/PET, (56 wt%) CCF/PET, (**b**) (56 wt%) CCF/PET, (0.1 wt%) CNT/CCF/PET, (1.0 wt%) CNT/CCF/PET, (2.0 wt%) CNT/CCF/PET.

**Figure 8 polymers-14-02892-f008:**
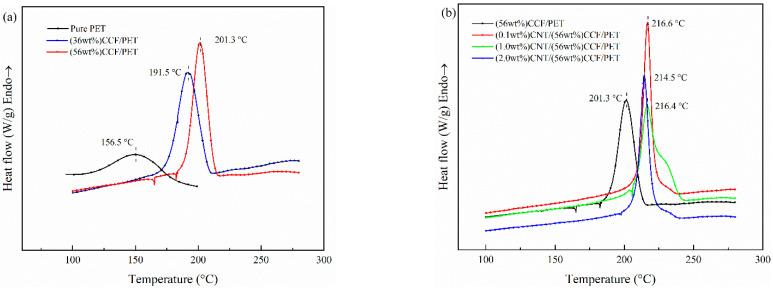
Crystallization behavior curves of pure PET, CCF/PET and CNT/CCF/PET composites: (**a**) pure PET, (36 wt%) CCF/PET, (56 wt%) CCF/PET, (**b**) (56 wt%) CCF/PET, (0.1 wt%) CNT/CCF/PET, (1.0 wt%) CNT/CCF/PET, (2.0 wt%) CNT/CCF/PET.

**Figure 9 polymers-14-02892-f009:**
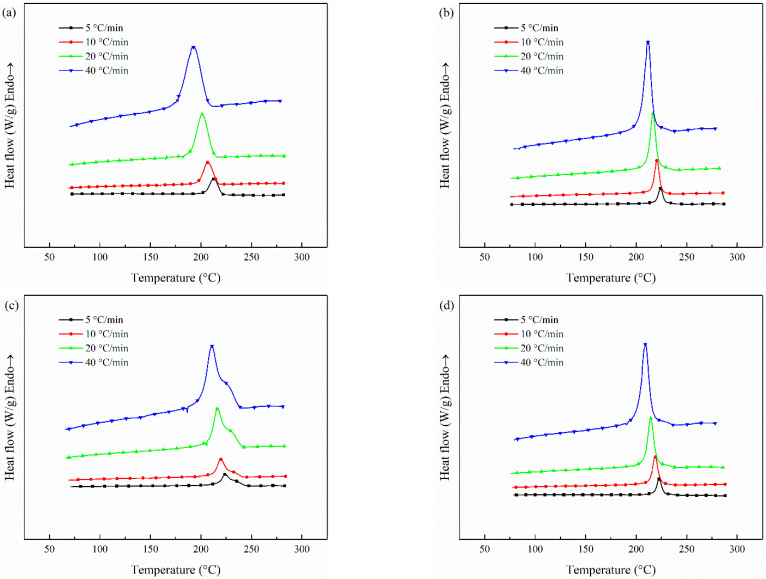
Non-isothermal melting crystallization curves of samples at different cooling rates: (**a**) (56 wt%) CCF/PET, (**b**) (0.1 wt%) CNT/CCF/PET, (**c**) (1.0 wt%) CNT/CCF/PET, (**d**) (2.0 wt%) CNT/CCF/PET.

**Figure 10 polymers-14-02892-f010:**
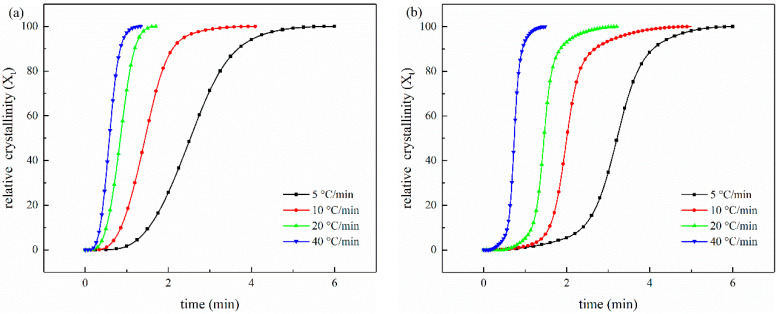
Relative crystallinity (*X_t_*) versus crystallization time (t/min) of non-isothermal melting crystallization of samples at different cooling rates: (**a**) (56 wt%) CCF/PET, (**b**) (0.1 wt%) CNT/CCF/PET, (**c**) (1.0 wt%) CNT/CCF/PET, (**d**) (2.0 wt%) CNT/CCF/PET.

**Figure 11 polymers-14-02892-f011:**
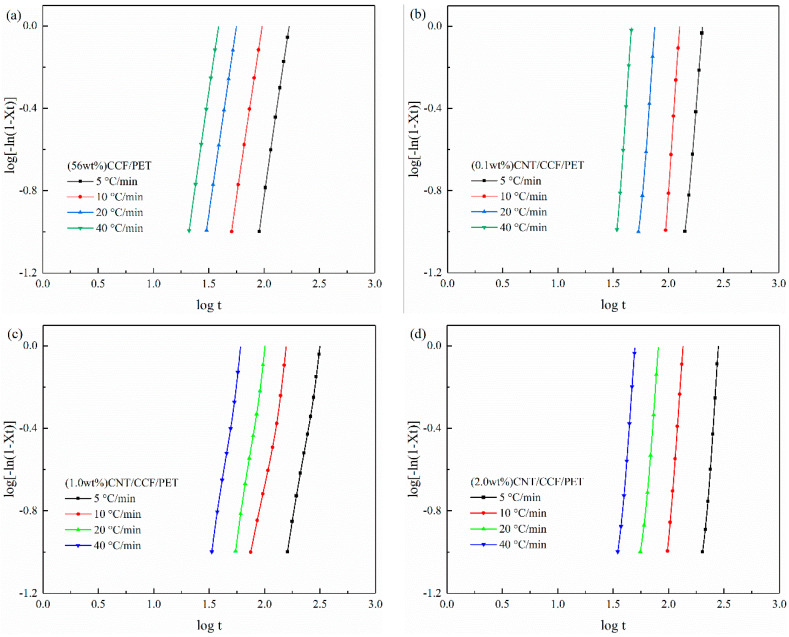
Avrami plots of *log[*−*ln(1−Xt)]* versus *log t* at different cooling rates: (**a**) (56 wt%) CCF/PET, (**b**) (0.1 wt%) CNT/CCF/PET, (**c**) (1.0 wt%) CNT/CCF/PET, (**d**) (2.0 wt%) CNT/CCF/PET.

**Figure 12 polymers-14-02892-f012:**
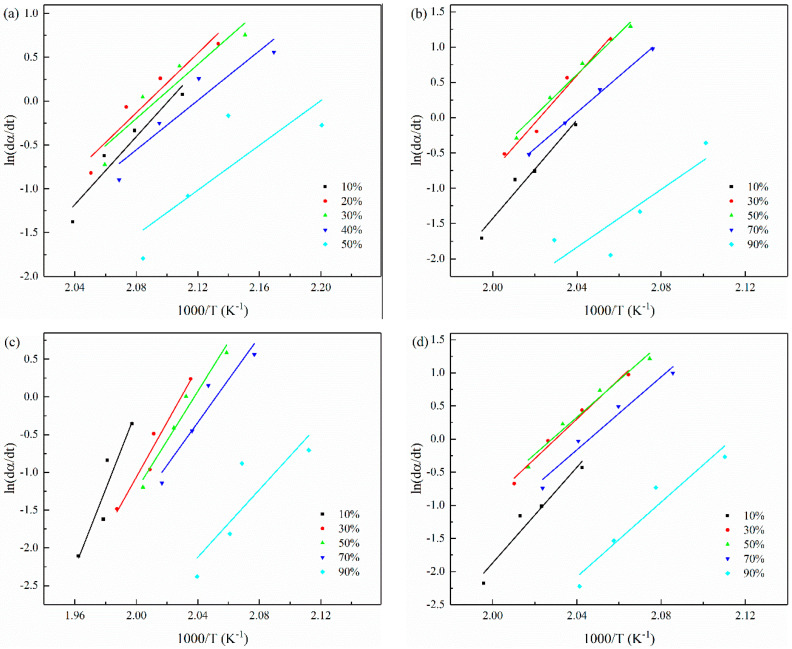
Friedman plots constructed from non-isothermal crystallization rate data: (**a**) (56 wt%) CCF/PET, (**b**) (0.1 wt%) CNT/CCF/PET, (**c**) (1.0 wt%) CNT/CCF/PET, (**d**) (2.0 wt%) CNT/CCF/PET.

**Table 1 polymers-14-02892-t001:** CNT/PET mass ratio of composite samples.

Samples	Matrix (PET) (wt%)	Antioxidant (wt %)		Carbon Nanotube (wt%)
		1010	168	
PET	99.6	0.2	0.2	—
CPET 1#	99.5	0.2	0.2	0.1
CPET 2#	99.0	0.2	0.2	0.5
CPET 3#	98.6	0.2	0.2	1.0
CPET 4#	97.6	0.2	0.2	2.0

**Table 2 polymers-14-02892-t002:** The formula of the prepreg tape.

Samples	Fiber Content (wt%)	CNT Content (wt%)
CCPET 1#	36	0
CCPET 2#	56	0
CCPET 3#	56	0.1
CCPET 4#	56	1.0
CCPET 5#	56	2.0

**Table 3 polymers-14-02892-t003:** Tensile properties of CNT/CCF/PET composites with different fiber content and CNT content.

Samples	Tensile Strength (MPa)	Tensile Modulus (GPa)	Failure Strain (%)
CCPET 1#	998.6 ± 29.5	11.8 ± 0.31	1.010
CCPET 2#	1567.1 ± 83.6	19.8 ± 0.27	0.850
CCPET 3#	1668.6 ± 56.8	20.9 ± 0.36	0.895
CCPET 4#	1728.7 ± 53.5	25.1 ± 0.22	0.721
CCPET 5#	1580.5 ± 72.4	21.9 ± 0.39	0.731

**Table 4 polymers-14-02892-t004:** Melting temperatures, melting enthalpy and relative crystallinity of pure PET, CCF/PET, and CNT/CCF/PET composites.

Samples	*T_m_* (°C)	∆*H_m_* (J/g)	*X_c_* (%)
Pure PET	242.6	—	—
CCPET 1#	248.7	24.79	17.7
CCPET 2#	249.1	23.14	16.5
CCPET 3#	248.6	26.32	18.8
CCPET 4#	248.5	28.14	20.1
CCPET 5#	248.8	21.15	15.1

**Table 5 polymers-14-02892-t005:** This is a table. Tables should be placed in the main text near to the first time they are cited.

Samples	*T_o_* (°C)	*T_c_* (°C)
Pure PET	242.6	—
CCPET 1#	248.1	24.79
CCPET 2#	249.1	23.14
CCPET 3#	247.6	24.92
CCPET 4#	248.5	25.19
CCPET 5#	248.8	21.15

**Table 6 polymers-14-02892-t006:** Non isothermal crystallization kinetic parameters based on Jeziorny modified Avrami equation.

Samples	*Φ*/°C min^−1^	*n*	*Z_c_*	*t_1/2_*/s
(56 wt%) CCF/PET	5	3.67	0.002	153.28
	10	3.59	0.194	86.43
	20	3.66	0.478	50.89
	40	3.71	0.712	35.26
(0.1 wt%) CNT/CCF/ PET	5	6.50	0.001	193.55
	10	6.87	0.021	112.48
	20	7.02	0.218	72.05
	40	7.72	0.476	44.38
(1.0 wt%) CNT/CCF/ PET	5	6.59	0.001	276.33
	10	6.89	0.033	130.38
	20	7.09	0.209	78.07
	40	7.36	0486	47.53
(2.0 wt%) CNT/CCF/ PET	5	3.24	0.023	291.75
	10	3.39	0.164	182.57
	20	3.53	0.439	93.68
	40	3.62	0.686	56.79

**Table 7 polymers-14-02892-t007:** The effective activation energy of CNT/CCF/PET with different CNT content by Friedman equation.

	*α*(%)	10	30	50	70	90
Samples		Effective Activation Energy (kJ/mol)				
56 wt% CCF/PET		−161.20	−141.17	−128.20	−117.22	−106.08
0.1 wt% CNT/CCF/PET		−289.57	−281.01	−242.35	−213.08	−170.10
1.0 wt% CNT/CCF/PET		−431.08	−302.71	−271.95	−235.95	−185.56
2.0 wt% CNT/CCF/PET		−299.80	−249.00	−232.37	−228.61	−233.87

## Data Availability

The data used in this research have been properly cited and reported in the main text.

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
