# Peer review of "Mechanical Properties, Melting and Crystallization Behaviors, and Morphology of Carbon Nanotubes/Continuous Carbon Fiber Reinforced Polyethylene Terephthalate Composites"

_polymers, 2022, doi:10.3390/polym14142892_

Round 1
Reviewer 1 Report
Recommendation: Major Revision
This paper has investigated the mechanical properties, melting and crystallization behaviors and morphology of carbon nanotubes/continuous carbon fiber reinforced polyethylene terephthalate composites. The main problem of this paper is the lack of in-depth discussion and related literature surveys. In addition, many important contributions in the related topic is completely missing in this manuscript. In this regard, what is the new information and novelty with this paper is questionable! Probably the paper can be considered but after substantial revision with more discussions. The more specific comments are given here.
1. The authors should include proper citations in each sentence, and please revise the reference format to MDPI style.
2. From line number 51st to 62nd, the authors argued that the incorporation of carbon nanotube (CNT) can improve the mechanical properties of the polymeric composites. However, the CNT-agglomerates which is caused by the poor disperse can decrease the mechanical properties. For these reasons, many literatures investigated the novel method for dispersing the CNT into the polymeric composites. The reviewer recommends to include more literatures to explain the different dispersion methods and compare them with the proposed method.
3. Please review the following reference “Facile Synthesis of Sprayed CNTs Layer-Embedded Stretchable Sensors with Controllable Sensitivity, Polymers (Basel). 13 (2021) 1–6. doi.org/10.3390/polym13020311”, “Electrical Stability and Piezoresistive Sensing Performance of High Strain-Range Ultra-Stretchable CNT-Embedded Sensors, Polymers (Basel). 14 (2022)”, and “Design of a highly flexible and sensitive multi-functional polymeric sensor incorporating CNTs and carbonyl iron powder, Compos. Sci. Technol. 207 (2021) 108725. doi:10.1016/j.compscitech.2021.108725”, to discuss the novel dispersion methods, and include them if they are required.
4. The percolation threshold is important factor when the CNTs are used in polymeric composites. Since, it affects the mechanical and electrical properties of the CNT-polymeric composites. Thus, please included the percolation threshold results in the present study.
5. The authors investigated the mechanical properties of the fabricated samples exposed to elevated temperatures. However, the proper reasons that the high mechanical properties are needed in high temperatures are not mentioned in the manuscript. Thus, the purpose of these experiments are curios. Please include the proper purpose and discussions of such experiments; and also please include the target applications of these fabricated samples.
6. The authors are recommended to revise all the figures to more scientific format considering the color, font, and size of the figures.
Reviewer 2 Report
The paper presents a study about the development of carbon nanotubes/continuous carbon fiber reinforced polyethylene terephthalate composites. At first glance, the study looks simple mixing of carbon nanotubes in carbon fiber reinforced polyethylene terephthalate composites to see what could happen with CF PET composites. The study overall (methods, materials, data treatment, explanations, writing style, the flow of thought, scientific knowledge, statistical analysis, and so on) is high quality. The manuscript is recommended for publication after revision.
1. The introduction is clear and concise, but the interest of the study is too general. The relevance of the study could be improved by specifying the sought performances and the kind of exigent applications.
2. Add Tensile modulus in Table 3 for CCPET 3# sample.
3. The Flexural and Impact properties must be studied before publication.
4. Some typo and grammar mistakes are observed. The manuscript should be proofread.
5. Why do DSC peaks move to higher temperatures?
6. Figure 9C, shows a shoulder peak? Author must justify this?
Round 2
Reviewer 1 Report
The authors have revised the manuscript considering the reviewer's comments. Thus, I think it can be published in this journal.
Reviewer 2 Report
Changes are made